# Double-Edged Sword Effect of Pyroptosis: The Role of Caspase-1/-4/-5/-11 in Different Levels of Apical Periodontitis

**DOI:** 10.3390/biom12111660

**Published:** 2022-11-08

**Authors:** Zhiwu Wu, Mingming Li, Xiaolin Ren, Rui Zhang, Jinfeng He, Li Cheng, Ran Cheng, Tao Hu

**Affiliations:** State Key Laboratory of Oral Diseases, Frontier Innovation Center for Dental Medicine Plus, National Clinical Research Center for Oral Diseases, Department of Preventive Dentistry, West China Hospital of Stomatology, Sichuan University, Chengdu 610041, China

**Keywords:** apical periodontitis, pyroptosis, caspase-1/-4/-5/-11, bone loss

## Abstract

The study was to investigate the effect of canonical and noncanonical pyroptosis in apical periodontitis. Proteins’ profiles of human apical periodontitis tissue were analyzed by label-free proteomics. Immunofluorescence was used to detect proteins related to pyroptosis in human apical periodontitis tissues and experimental apical periodontitis models. A dual experimental apical periodontitis model with both smaller (mandible) and larger (maxilla) bone lesions was established. THP-1-derived macrophages were stimulated with *P. gingivalis* lipopolysaccharide in vitro with or without the caspase-1/-4/-5 inhibitor Ac-FTDL-CMK. Propidium iodide staining, lactic dehydrogenase release and Western blot were applied to evaluate cell death and the protein expression. Caspase-1/-4/-5 were expressed in human apical periodontitis tissues. Caspase-1/-11 were involved in bone loss in experimental apical periodontitis. Caspase-1/-11 inhibitors reduced bone loss in larger lesions (maxilla) but accelerated bone loss in smaller lesions (mandible). Caspase-1/-4/-5 inhibitors also showed double-edged sword effects on propidium iodide staining and lactic dehydrogenase release in vitro. The expression of cleaved-caspase-1/-4/-5, mature interluekin-1β and gasdermin D N-terminal domain increased in THP-1-derived macrophages after lipopolysaccharide stimulation but decreased after treatment with Ac-FTDL-CMK. Pyroptosis contributed to apical periodontitis and excited a double-edged sword effect in inducing bone loss in vivo and cell death in vitro.

## 1. Introduction

Apical periodontitis (AP) is defined as inflammation that affects the periapical region of a tooth root [1,2]. It is mainly caused by multiple microbes that originate from the root canal through the apical foramen. Obligate anaerobes, such as *Porphyromonas gingivalis* (*P. gingivalis*), *Porphyromonas endodontalis*, *Fusobacterium nucleatum* and *Tannerella forsythia*, are the dominant pathogenic species and initiate a host complex immune response in AP [3,4,5,6]. The formation and progression process of AP is dynamic [2]. This dynamic equilibrium between anti- and proinflammatory processes is affected by intricate interactions between inflammatory cells (macrophages, neutrophil, mast cell, lymphocytes, T cell, etc.), cytokines, chemokines, RANK/RANKL/OPG system and neuropeptides [2,7], which influence the balance between tissue destruction and regeneration in AP [7]. Innumerable investigations have been conducted to study the pathogenesis of AP and this dynamic shift; however, the exact etiological factors and the mechanisms associated with molecular changes has been not well elucidated [8,9]. What is the crucial factor in mediating this homeostasis in a periapical lesion?

Pyroptosis, a form of inflammatory programmed cell death mediated by gasdermin family proteins composed of gasdermin A/B/C/D/E (GSDMDA/B/C/D/E) and DFNB59 (Pejvakin, PJVK) [10,11], caspase-1 and caspase-4/-5/-11 (caspase-11 is homologous to human caspase-4/-5 in murine), which are inflammatory caspases that induce canonical and noncanonical pyroptosis when activated, respectively [12]. Caspase-1 is activated by various inflammasomes that recognize pathogenic signals [13], while caspase-4/-5/-11 are directly activated upon recognition of cytosolic lipopolysaccharide (LPS) [14,15]. Activated caspase-1/-4/-5/-11 cleave GSDMD; subsequently, the gasdermin-N domain (GSDMD-N) forms membrane pores to trigger pyroptosis with mature interleukin-1β (IL-1β) release [16,17].

Macrophages, which are innate immune cells, defend against invading microbes by ingesting and destroying them [18], and then releasing cytokines such as IL-1β and tumor necrosis factor (TNF)-α [7], thus contributing to inflammatory responses and bone loss [19]. Microbial infections are associated with pyroptosis in macrophages [20]. The activation of the inflammasome and pyroptosis were detected when macrophages were stimulated by microbial stimuli and oxidative stress [21]. Consequently, pyroptosis in macrophages may be involved in the pathogenesis of diseases [22], including AP, in which macrophages infiltrate [23,24].

In recent years, the dual effects of pyroptosis were reported, especially in the field of tumor immunity. Cytokines produced from pyroptosis accelerate the development of tumors by evading immune surveillance but recruit immune cells to resist the tumor [25]. Pyroptosis can not only inhibit tumor cell growth, but also change a microenvironment suitable for tumor cell proliferation [26]. Thus, what is the effect of pyroptosis in AP?

The current studies focus on the role of the canonical pyroptosis pathway in AP. There are few studies concerning the noncanonical pyroptosis pathway. Caspase-4/-5/-11, which are molecules related to the noncanonical pyroptosis pathway, cannot be ignored in the production and regulation of inflammation [12]. Will canonical and noncanonical pyroptosis be involved in AP, how do they affect the initiation and progression of AP and are they protective or detrimental? The aim of this study was to investigate the role of caspase-1- and caspase-4/-5/-11-induced pyroptosis in AP by using in vivo and in vitro models.

## 2. Material and Methods

### 2.1. Clinical Specimens

Human apical periodontitis tissues (HAPT, a total number of 23) from healthy human donors (aged 18 to 64 years) were obtained from extracted human teeth or teeth that underwent periapical surgery at West China Hospital of Stomatology (Sichuan University, Chengdu, China). The inclusion criteria were that teeth were diagnosed as chronic AP with periapical radiolucent lesions and required periapical surgery or extraction. The exclusion criteria were patients with malignant tumors or autoimmune diseases, and patients who smoked or had antibiotic use for at least 3 months. In addition, human periodontal ligaments (HPDL, a total number of 47) were collected from healthy nonsmoking donors (aged 14 to 46 years) after premolar extraction for orthodontic reasons and third molar extraction.

### 2.2. Label-Free Proteomics

Clinical species were collected for detecting proteins by label-free proteomics. More label-free proteomics details can be found in Appendix A.

### 2.3. Establishment of the Experimental Apical Periodontitis (EAP) Rat Model

Female Sprague-Dawley (SD) rats aged 12–16 weeks and weighing 220 g were purchased from Beijing Vital River Laboratory Animal Technology Co., Ltd. The pulp of the bilateral upper and lower first molars was exposed to the oral environment using a #1/4 dental round bur [27]. Rats in each group were anaesthetized after 1~4 weeks. Caspase-1 inhibitor VX765 [27] (50 mg/kg; Biochempartner, Hangzhou, China) and caspase-11 inhibitor Wedelolactone [28] (Wed, 4 mg/kg; Biochempartner) dissolved in 20% cremophor (Sigma-Aldrich, Shanghai, China) were intraperitoneally injected once a day for 14 consecutive days since week 1 [29] (caspase-1/11 was activated in apical region at week 1). All rats were anaesthetized at 4 weeks after pulp exposure, and the mandibles and maxilla were fixed with 4% paraformaldehyde at 4 °C for 72 h. Next, the mandibles and maxilla were stored in 75% ethanol prior to microcomputed tomography (micro-CT) scanning. Then, the specimens were rinsed and decalcified with 10% ethylenediaminetetraacetic acid for 12 weeks, dehydrated and embedded in paraffin. Serial sections of 5 μm thickness were cut in the mesiodistal direction for further experiments [30]. More experimental details can be found in Appendix A.

### 2.4. Micro-CT Analysis

The mandibles and maxillary were placed in an airtight cylindrical sample holder. A Scanco μCT50 imaging system (Scanco Medical AG, Bassersdorf, Zurich, Switzerland) was used to assess the periapical bone loss of the distal root of the first mandibular molar and the disto-buccal root of the first maxillary molar. The scanning was performed at 70 kVp and 200 μA, with a resolution of 10 μm and 300-millisecond exposure time. The segment selection method was performed as previously described [27]. Bone volume relative to the total volume (BV/TV), trabecular number (Tb. N), trabecular thickness (Tb. Th) and trabecular space (Tb. Sp) were counted for quantification to evaluate bone loss of periapical regions [31].

### 2.5. Hematoxylin-Eosin Staining (HE) and Immunofluorescence

The slice of EAP that exhibited a clear root canal apex and clinical specimens were selected for staining. HE was performed according to the standard procedure (Biosharp Life Science, Hefei, China).

The immunoreaction sequences were stained by anti-caspase-1, anti-caspase-4, anti-caspase-5, anti-caspase-11 and anti-CD68. The nuclei were visualized by using Hoechst 33342 (Hoechst; ImmunoChemistry Technologies. LLC, Davis, CA, USA). Images were taken by a fluorescence microscope (Leica DM2000, Leica Corporation, Weztlar, Germany). The relative intensity and colocalization of two proteins [Pearson’s correlation coefficient (PCC), overlap coefficient (OC) and scatterplot] were assessed by ImageJ software.

Primary antibody: mouse anti-caspase-1 (sc-398715, Santa Cruz Biotechnology Inc. Dallas, CA, USA), mouse anti-caspase-5 (sc-393346, Santa Cruz Biotechnology); mouse anti-caspase-11 (sc-374615, Santa Cruz Biotechnology); mouse anti-CD68 (ab201340, Abcam, Cambridge, UK); rabbit anti-caspase-1 (ab1872, Abcam); rabbit anti-caspase-4 (ab25898, Abcam); rabbit anti-caspase-5 (ab40887, Abcam).

Secondary antibody: 488 goat anti-mouse IgG (A11011, Invitrogen, Carlsbad, CA, USA), 488 goat anti-rabbit IgG (A11008, Invitrogen), 555 goat anti-mouse IgG (A21422, Invitrogen) and 555 goat anti-rabbit IgG (A21428, Invitrogen).

### 2.6. TUNEL Staining

The procedure was performed according to the instructions of the DeadEnd™ Fluorometric TUNEL System (Promega Co., Ltd., Madison, WI, USA). All nuclei were visualized by using Hoechst 33,342. Images were taken by a fluorescence microscope. The TUNEL-positive rate was calculated, PCC, OC and scatterplot (the relative and colocalization of TUNEL and caspase-4/-5/-11 (anti-caspase-4: ab25898, Abcam; anti-caspase-5: ab40887, Abcam; and anti-caspase-11: sc-374615, Santa Cruz Biotechnology)) were assessed by ImageJ [32,33].

### 2.7. Cell Culture

THP-1 cells (ATCC^®^ TIB-202™) were purchased from the American Type Culture Collection. The cell line was identified and considered to be “identical” to the reference cell line in the Cell Bank STR database, as the STR profile yields a 100% match (GENEWIZ, Inc., Suzhou, China). Cells were cultured in RPMI-1640 (Gibco, Grand Island, NY, USA) containing 10% fetal calf serum (Biowest, Nuaillé, France), 100 U/mL penicillin and 100 μg/mL streptomycin (HyClone, UT, USA) and incubated at 37 °C in 5% CO_2_. To obtain THP-1-derived macrophages, THP-1 cells were seeded at a density of 1 × 10^7^ cells per well in a six-well culture plate and stimulated with 100 ng/mL Phorbol 12-Myristate 13-Acetate (PMA) (Sigma-Aldrich) for 24 h, followed by further incubation in an RPMI medium in the absence of PMA for 24 h [34].

### 2.8. Lactic Dehydrogenase (LDH) Release Assay

THP-1-derived macrophages were serum-starved for 24 h followed by treatment with 0.1~10 μg/mL *P. gingivalis* LPS (Invivogen, San Diego, CA, USA) and/or 10 μM caspase-1/-4/-5 inhibitor, Ac-FLTD-CMK (Selleck Chemicals, Houson, TX, USA) [35], at 5% CO_2_ for 6 h. A cell culture supernatant was collected, and the LDH release was detected using a CytoTox 96 Non-Radioactive Cytotoxicity Assay Kit (Promega) according to the manufacturer’s protocol. The experiments were carried out on three independent experiments. All values represent the percentage of LDH release compared with a maximum LDH release control (Lysis Solution). The absorbance was measured using a microplate meter (SpectraMax iD3, Molecular Devices, LLC, San Jose, CA, USA).

### 2.9. Propidium Iodide (PI) Staining

THP-1-derived macrophages were serum-starved for 24 h followed by treatment with 0.1~10 μg/mL *P. gingivalis* LPS and/or 10 μM caspase-1/-4/-5 inhibitor, Ac-FLTD-CMK, at 5% CO_2_ for 6 h. To assess cell death, propidium iodide (PI) marks dying cells, while 496-diamidino-2-phenylindole (DAPI) stains all nuclei. Images were captured by using a fluorescence microscope (Olympus IX73, Olympus Corporation, Tokyo, Japan). The experiments were either carried out in triplicate or quadrupled in three independent experiments. The PI positive cells were calculated by Image J.

### 2.10. Western Blot

THP-1-derived macrophages were serum-starved for 24 h followed by treatment with 1 μg/mL *P. gingivalis* LPS and/or 10 μM caspase-1/-4/-5 inhibitor Ac-FLTD-CMK at 5% CO_2_ for 6 h. Cells were collected, and proteins were extracted using a Total Protein Extraction Kit (Signalway Antibody, Greenbelt, MD, USA) according to the manufacturer’s instructions. Equal amounts of protein were separated using 10% SDS–PAGE (Bio–Rad) and transferred to polyvinylidene difluoride membranes (GE Healthcare Life Science, Pittsburgh, PA, USA). The membrane was incubated in a blocking buffer (5% BSA in Tris-buffered saline containing 0.1% Tween 20) at room temperature for 1 h and then incubated with the following primary antibodies overnight at 4 °C: anti-caspase-1 (sc-398715, Santa Cruz Biotechnology), anti-caspase-4 (sc-56056, Santa Cruz Biotechnology), anti-caspase-5 (sc-393346, Santa Cruz Biotechnology), anti-GADMD-N (ab215203, Abcam), anti-IL-1β (ab9722, Abcam) and anti-GAPDH (Signalway Antibody). After washing, the membranes were incubated with horseradish peroxidase-conjugated secondary antibody for 1 h at 37 °C. A chemiluminescence kit (Bio-Rad, Hercules, CA, USA) was used to visualize the immunoreactive bands. The bands were analyzed by ImageJ software.

### 2.11. Data Analysis

All data were presented as the mean ± S.D. The statistically significant difference among groups was assessed using Student’s *t* test or a one-way ANOVA with SPSS 22.0 (IBM Corp. New York, NY, USA). Comparisons between two groups were performed by Student’s *t* test. For more than two groups, a one-way ANOVA was performed. *p* < 0.05 was considered statistically significant.

## 3. Results

### 3.1. Caspase-1 and Its Related Proteins Were Only Detected in Human Chronic AP

A total of 8 samples of AP and 32 samples of periodontal ligament tissue were collected, which mixed into 4 samples, respectively, for label-free proteomics due to individual differences and samples’ weight [36]. There were 418 proteins observed to undergo significant changes in abundance (SCA), including 121 downregulated proteins and 297 upregulated proteins. Moreover, there were obvious differences between the AP and periodontal ligament tissue. The consistent presence/absence expression profile (CEP) showed that 336 differentially expressed proteins (DEPs) were only detected in AP samples and 37 DEPs were only detected in periodontal ligament samples (Figure 1A; Appendix A).

Subsequently, the proteins possibly related to pyroptosis were analyzed by using String (https://cn.string-db.org/ accessed on 22 May 2022) to structure protein–protein interactions (PPIs) (Figure 1B). Among the 28 proteins possibly related to pyroptosis, 12 proteins were detected in our study. The PPI including these 12 proteins was established (Figure 1C). Among the 12 proteins, 6 had significant differences, and 4 were only detected in AP. Caspase-1, apoptosis-associated speck-like protein containing a CARD (PYCARD) and eukaryotic translation initiation Factor 4 gamma 2 (EIF4G2) were detected in the AP tissue. The ATP-dependent RNA helicase DDX3X (DDX3X) (*p* = 0.0232) and eukaryotic initiation Factor 4A–I (EIF4A1) (*p* = 0.0011) were upregulated, but ATP-dependent RNA helicase A (DHX9) (*p* = 0.0149) and polypyrimidine tract-binding protein 1 (PTBP1) (*p* = 0.0495) were downregulated in the AP tissue compared to periodontal ligament tissue (Figure 1D). It was found that most proteins were associated with caspase-1, which mediates canonical pyroptosis.

### 3.2. Caspase-1/-4/-5 Differentially Mediated Canonical and Noncanonical Pyroptosis in Human AP

Human periodontal ligament tissues (HPDL) and human apical periodontitis tissues (HAPT) were collected in this experiment. The results showed that caspase-1/-4/-5 were expressed to a great extent in HAPT compared to HPDL. Furthermore, the extent of caspase-1/-4/-5 was different in AP. Not all cells simultaneously expressed caspase-4 and caspase-5 (PCC = 0.696 ± 0.140; OC =0.817 ± 0.109) (Figure 2A and Appendix A). Cells co-expressed caspase-1 and caspase-5 (PPC = 0.773 ± 0.176; OC = 0.784 ± 0.166) but not caspase-1 and caspase-4 (PCC = 0.120 ± 0.015; OC = 0.203 ± 0.047) (Figure 2B,C and Appendix A). Caspase-4/-5 was positively stained in a large number of cells in the periapical lesion. Most caspase-4^+^ (PCC = 0.720 ± 0.249; OC = 0.814 ± 0.188) and caspase-5^+^ (PCC = 0.663 ± 0.181; OC = 0.761 ± 0.145) cells underwent cell death (Figure 2D,E and Appendix A). CD68^+^ macrophages expressed caspase-5 (PCC = 0.853 ± 0.091; OC = 0.861 ± 0.093), caspase-1 (PCC = 0.857 ± 0.019; OC = 0.902 ± 0.085) and a small amount of caspase-4 (PCC = 0.402 ± 0.022; OC = 0.482 ± 0.027) (Figure 2F,G and Appendix A). It was surmised that caspase-1-mediated canonical and caspase-4/-5-mediated noncanonical pyroptosis were involved in human AP.

### 3.3. Caspase-1/-11-Mediated Pyroptosis Contributed to the Progression of EAP

The EAP was established by drilling an opening in the mandibular first molar to reach the pulp to allow infection by oral bacteria in the pulp and periapical tissue. The apical lesion developed gradually, and some immune cells infiltrated in the periapical area (Figure 3A). Caspase-1 and caspase-11 were observed mildly in the first week after the establishment of the experimental AP rat model, but from the second week, the colocalization expression of caspase-1 and caspase-11 was easily found (Figure 3B and Appendix A). The great majority of caspase-11^+^ cells underwent cell death (Figure 3C and Appendix A), suggesting that caspase-11 may induce noncanonical pyroptosis in EAP. At week 1, the mild expression of CD68^+^ macrophages could be observed, but at 2 weeks, CD68^+^ macrophages were distinctly increased. A large proportion of CD68^+^ macrophages expressed caspase-1 and caspase-11 starting from 2 weeks and 3 weeks, respectively (Figure 3D,E and Appendix A). These results suggest that caspase-1 and caspase-11 mediate canonical and noncanonical pyroptosis in EAP.

Pyroptosis appeared in EAP as early as 1~2 weeks when apical lesions started to develop. Pyroptosis reached a high level after week 2. However, does pyroptosis influence the outcome of AP to improve bone loss or to protect AP from developing? For further investigation, an EAP model with both smaller and larger bone lesions was established.

### 3.4. Establishment of a Dual EAP Model with Both Smaller (in Mandible) and Larger (in Maxilla) Bone Lesions

The maxilla is more susceptive to EAP than the mandible [37], which provided evidence for us to establish a dual EAP model in which a different lesion was observed in the same rat. In the EAP, the maxillary molar and mandibular molar were selected to simulate larger and smaller periapical lesions, respectively (Figure 4A). The results suggest that the bone density around the maxillary molar was higher than that around the mandibular molar. The EAP induced bone loss around the apical site of the root. The lesion of the maxillary molar was largely changed compared to the change in the mandibular molar (Figure 4B).

### 3.5. Pyroptosis Is a Double-Edged Sword in Inducing Bone Loss in Periapical Inflammation

To evaluate the effect of pyroptosis, the dual EAP model was treated with the caspase-1 inhibitor VX765 [27,38] and/or the caspase-11 inhibitor Wedelolactone [28]. Compared to the control, the periapical lesion in the maxillary distobuccal root showed significant changes in BV/TV (*p* < 0.001), Tb. Th (*p* < 0.001), Tb. N (*p =* 0.004) and Tb. Sp (*p* < 0.001) (larger lesion). There were significant differences in Tb. Th (*p* = 0.016) in the mandibular distal root compared to the control (smaller lesion). In the VX765+Wed group, the maxillary distobuccal root (larger lesion) showed inhibitory effects on Tb. N (*p* < 0.001) and Tb. Sp (*p* = 0.007) compared to the AP group. Interestingly, VX765+Wed accelerated bone resorption in the mandibular distal root (smaller lesion). BV/TV (*p* = 0.018), Tb. Th (*p* = 0.029), Tb. N (*p* = 0.031) and Tb. Sp (*p =* 0.014) showed greater bone loss than the AP group (Figure 5). The expressions of RANKL, RANK and OPG were analyzed by IHC. The results showed that RANK (*p =* 0.010) and OPG (*p =* 0.011) increased in the AP of mandibular distal root (smaller lesion). VX765+Wed inhibited the expression of OPG (*p =* 0.019), but not RANK (Appendix A).

The bone loss was different between the maxilla and the mandible. The differences in the bone metabolism, bone density and ratio of trabecular bone to cortical bone may influence their sensitivities [37]. Their blood supplies also differed [39]. Considering that the same method of pulp exposure and duration were applied in vivo, the dual model in vitro was established by using a different concentration of LPS.

### 3.6. LPS Induced Pyroptosis in THP-1-Derived Macrophages. Inhibition of Pyroptosis Showed a Double-Edged Sword Effect in the In Vitro Models

It was shown that macrophages underwent pyroptosis, which was involved in the progression of AP in vivo. THP-1 cells are human monocytic leukaemia cells and have been widely used to induce macrophages and study their role and function in vitro [40]. Pyroptotic events result in LDH release and PI staining. LDH is released from lysed cells and is considered a specific indicator of pyroptosis. As PI can pass through pores that form on the cell membrane, PI staining indicated not only lysed cells but also intact cells [17].

THP-1-derived macrophages were stimulated with different levels of *P.g* LPS (0.1 μg/mL, 1 μg/mL, 10 μg/mL) and to simulate apical inflammation. Pretreated and nonpretreated Ac-FLTD-CMK were used to inhibit different durations of caspase activation. Surprisingly, different levels of LPS and different inhibition durations of Ac-FLTD-CMK (pretreated, nonpretreated) induced a double-edged sword effect in the LDH release [41]. Figure 6A reveals that a low concentration of LPS increased LDH release compared to the control. Pretreatment and nonpretreatment with Ac-FLTD-CMK inhibited LDH release. When the LPS concentration increased to 1 μg/mL, LDH increased after LPS stimulation. Nonpretreatment with Ac-FLTD-CMK inhibited the effect of LPS, but pretreatment with Ac-FLTD-CMK had no effect. When the concentration of LPS reached the highest 10 μg/mL, both nonpretreated and pretreated Ac-FLTD-CMK accelerated the effect of LPS. A longer duration of Ac-FLTD-CMK pretreatment enhanced the effect. PI staining also indicated a double-edged sword effect (Figure 6B). When the concentration of LPS was lower than 1 μg/mL, Ac-FLTD-CMK inhibited PI staining. The effect was the opposite when LPS was increased to 10 μg/mL. Interestingly, a high level of LPS did not induce correspondingly more cell death or pore formation, and pyroptosis seemed to protect the cells from death or injury. When PI staining reached the highest level in the 1 μg/mL LPS group, Ac-FLTD-CMK largely decreased PI staining. Then, the contraction of 1 μg/mL LPS was selected to detect the protein level.

Pro-caspase-1 (*p* = 0.001), pro-caspase-4 (*p* < 0.001), pro-caspase-5 (*p* = 0.001), cleaved-caspase-1 (*p* < 0.001), cleaved-caspase-4 (*p* = 0.005), cleaved-caspase-5 (*p* = 0.002), pro-IL-1β (*p* < 0.001), mature IL-1β (*p* < 0.001) and GSDMD-N (*p* < 0.001) were upregulated in the LPS group compared to the control group. In the LPS+Ac-FLTD-CMK group, cleaved caspase-1 (*p* < 0.001), cleaved caspase-4 (*p* = 0.008), cleaved caspase-5 (*p* = 0.001), GSDMD-N (*p* = 0.001) and mature IL-1β (*p* < 0.005) were downregulated compared to those in the LPS group. In the LPS+Ac-FLTD-CMK pretreatment group, cleaved caspase-1 (*p* = 0.004) and pro-IL-1β (*p* = 0.009) were downregulated significantly compared to those in the LPS group. However, cleaved caspase-1 (*p* = 0.046) and cleaved caspase-5 (*p* = 0.019) were upregulated compared to those in the LPS+Ac-FLTD-CMK group. This result indicated that Ac-FLTD-CMK (both untreated and pretreated) inhibited the production of GSDMD-N cleaved-caspase-1 and mature IL-1β. The activation of caspase-4 and caspase-5 was inhibited by nonpretreatment with Ac-FLTD-CMK but pretreatment with Ac-FLTD-CMK. These results showed that the duration of Ac-FLTD-CMK treatment affected the extent of caspase activation, which influenced pyroptosis accordingly (Figure 7).

## 4. Discussion

AP is an inflammatory disease within periapical tissues that is most frequently induced by poly-microbes originating from pulp [4]. Tissue destruction and bone loss are self-limiting in most cases of AP. Both protective and destructive immunoreactions take place during the interactions between host cells and pathogenic microbes. The dynamic shift between anti- and proinflammatory processes influences tissue destruction or repair during the progression of AP. The equilibrium between destructive and protective processes is driven by pleiotropic regulatory mechanisms, including immune cells and inflammatory mediators [7]. The study indicated that pyroptosis might also participate in protective and destructive processes.

The proteomics results showed that a group of proteins related to caspase-1 was increased in the AP tissue. PPI showed that caspase-1 was directly associated with 7 proteins, which proved the importance of caspase-1 in AP. However, caspase-4/-5 were not detected in label-free proteomics, which may be linked to the sensitivity of label-free proteomics. These were factors, such as signal overlap of coeluting peptides, background noise and reduced sampling rate of the eluding peptides, that interfered with protein identification and quantification [42,43]. Therefore, this study investigated the role of both canonical and noncanonical pyroptosis in AP.

The activation of caspase-1 depends on canonical inflammasome, whereas caspase-4/-5 can recognize intracellular bacterial LPS directly and bind the lipid A motif of LPS, forming noncanonical inflammasome. Activated caspase-4/-5 cleave GSDMD and IL-1β, which leads to noncanonical pyroptosis and triggers secondary activation of canonical inflammasome [44]. In this study, most caspase-5^+^ cells and caspase-4^+^ cells underwent cell death, which indicated that caspase-4/-5-mediated noncanonical pyroptosis may be vital in human AP. Macrophages, the major effector cells of innate immunity, play important roles in host defense. In human AP, macrophages undergo caspase-1/-5-mediated pyroptosis, while caspase-4 is rarely observed. The in vitro experiment showed that caspase-1, caspase-4 and caspase-5 were all activated in THP-1-derived macrophages after stimulation with *P. gingivalis* LPS, which was different from the results in human AP tissue. It was reported that the Shigella effector OspC3 can bind to caspase-4 P19, thus blocking the activation of caspase-4 in the adaptive intracellular environment [45]. The activation of caspase-4 was inhibited by the bacterial type III secretion system (T3SS) effector NleF in intestinal epithelial cells [46]. Hence, it was thought that the activation of caspase-4 could be changed by other bacterial effectors in chronic AP due to the mixed microflora of infected root canals being involved in the pathology of AP [47].

Caspase-11 is a murine homologue of human caspase-4/-5 [15]. Caspase-11 is activated when it directly recognizes LPS and is activated, subsequently cleaving GSDMD and leading to pyroptosis such as caspase-4/-5 [48]. In EAP, not only caspase-1-mediated canonical pyroptosis but also caspase-11-mediated noncanonical pyroptosis induce periapical inflammation.

Pyroptosis is closely related to bone-related diseases. Pyroptosis promotes bone loss in osteoporosis [49]. Additionally, the overexpression of the NLRP3 inflammasome aggravates inflammatory osteolysis, causing arthritis, osteomyelitis and periodontal disease [11]. Pyroptosis is also involved in bone loss in the periapical region [27]. Studies have indicated that pyroptosis aggravates bone loss in most reported diseases. It was also reported that pyroptosis and inflammasomes act as double-edged swords with both protective and detrimental effects for host defense and bone remodeling [50]. How and when did the protective or detrimental effects happen? We established an in vivo EAP model and an in vitro model to investigate the protective and detrimental effects.

There was a difference in susceptibility to AP between the maxilla and mandible. It was reported that the maxilla was more prone to AP than the mandible in terms of microstructural changes in the trabecular bone [37]. For the in vivo EAP model, the maxillary molar and mandibular molar were selected to simulate smaller and larger inflammation, respectively. The results showed that bone loss in the distobuccal root of the maxilla was more obvious than that in the distal root of the mandible.

To further determine the effect of pyroptosis in AP, caspase-1 and caspase-11 inhibitors were used. Interestingly, an opposite trend of bone loss in the maxilla and mandible was observed. In brief, bone loss increased when caspase-1/-11 were inhibited in the case of a smaller lesion (mandible), but in the maxilla, where bone lesions are more severe, bone loss was inhibited with caspase-1/-11 inhibitors. OPG, which limited osteoclastogenesis by blocking the interactions of RANKL with RANK, was also found to be decreased by caspase-1/-11 inhibition in a smaller mandibular lesion. It might contribute to the increased bone loss in the mandible. Therefore, the results suggest that pyroptosis acts as a double-edged sword, conferring both a protective and detrimental potential for bone remodeling. When a bone lesion is smaller, pyroptosis is protective to prevent excessive bone loss. However, when bone loss is larger, pyroptosis plays a detrimental role during the process. Indeed, appropriate inflammasomes are required for bone homeostasis, while excessive inflammasomes contribute to the progression of diseases [50,51,52].

Macrophages were found to be involved in human AP and EAP. Some studies have found that caspase-1 and caspase-4 can be activated by *P. gingivalis*, *Treponema denticola* and *Tannerella forsythia* stimulation in THP-1-derived macrophages [53]. *P. gingivalis*, the main pathogenic bacteria of periodontitis, is also abundant in infected root canals [54,55], so it is also considered one of the main pathogenic bacteria of AP [56]. LPS is the main virulence factor of *P. gingivalis* [57], which can simulate inflammatory stimuli in the study of AP in vitro [27,58]. Therefore, *P. gingivalis* LPS was used to stimulate THP-1-derived macrophages to investigate the role of pyroptosis.

Protective and detrimental effects were also detected in vitro. The double-edged sword effect presented when the concentration of LPS and the duration of Ac-FLTD-CMK changed in LDH release and PI staining. When cells were stimulated with different levels of LPS, cell death always appeared and was maintained at a certain level. Interestingly, when THP-1 cells were stimulated with high levels of LPS, Ac-FLTD-CMK increased cell death, suggesting that pyroptosis was protective in these circumstances. When moderate or mild levels of LPS were applied, Ac-FLTD-CMK inhibited or maintained cell death. This might be the reason why pyroptosis was maintained at a certain level with stimulation with 0.1~10 μg/mL LPS. PI staining indicated pore formation on the cell membrane. Pyroptosis was also protective when stimulated with 10 μg/mL LPS. When PI staining was the highest in 1 μg/mL LPS, Ac-FLTD-CMK strongly inhibited the pyroptotic effect. This result suggested that pyroptosis was protective when the stimulation was severe. When the stimulation weakened but pyroptosis persisted, the effect became destructive.

Finally, pyroptosis-related proteins were detected. A typical concentration of 1 μg/mL LPS with pretreated or untreated Ac-FLTD-CMK was selected. The activation of IL-1β and GSDMD-N was inhibited by both Ac-FLTD-CMK treatments, suggesting that Ac-FLTD-CMK successfully decreased LPS-induced pore formation. Even when the pretreatment groups had a longer duration of inhibition, the activations of caspase-1 and caspase-5 were not reduced compared to those in the untreated group. Correspondingly, LDH release in the pretreatment group was also not reduced, suggesting that the extent of cell death was not inhibited. In all, the inhibition of caspases decreased the level of GSDMD-N, but this may not decrease cell death accordingly. The finding was partly accordant to a report that macrophages underwent two statuses: cell hyperactivation to release IL-1β (alive cells) or pyroptosis (dead cells) when elicited by inflammasomes [41].

AP is the prevalent oral disease, and it brings a hidden burden worldwide [59]. Root canal therapy is a fundamental and preferred method of treating AP nowadays. However, the outcome of root canal therapy may be influenced by the host immune response [60] and the complexity of the root canal system [61]. The rational use of this dual effect for us will be conducive to further exploring the mechanisms of immunity in AP, and to provide ideas for patients to develop new treatment strategies based on pyroptosis.

## 5. Conclusions

Caspase-1-related proteins were highly expressed in AP tissue compared with normal tissue. Caspase-1-mediated canonical and caspase-4/-5-mediated noncanonical pyroptosis are involved in AP. Pyroptosis is a double-edged sword in inducing bone loss in vivo. It is detrimental when the bone lesion is larger and protective when bone loss is smaller. The in vitro study verified the double-edged sword effects. Pyroptosis is protective when stimulated with high levels of LPS. The effect becomes destructive when the concentration of LPS decreases (Figure 8).

## Figures and Tables

**Figure 1 biomolecules-12-01660-f001:**
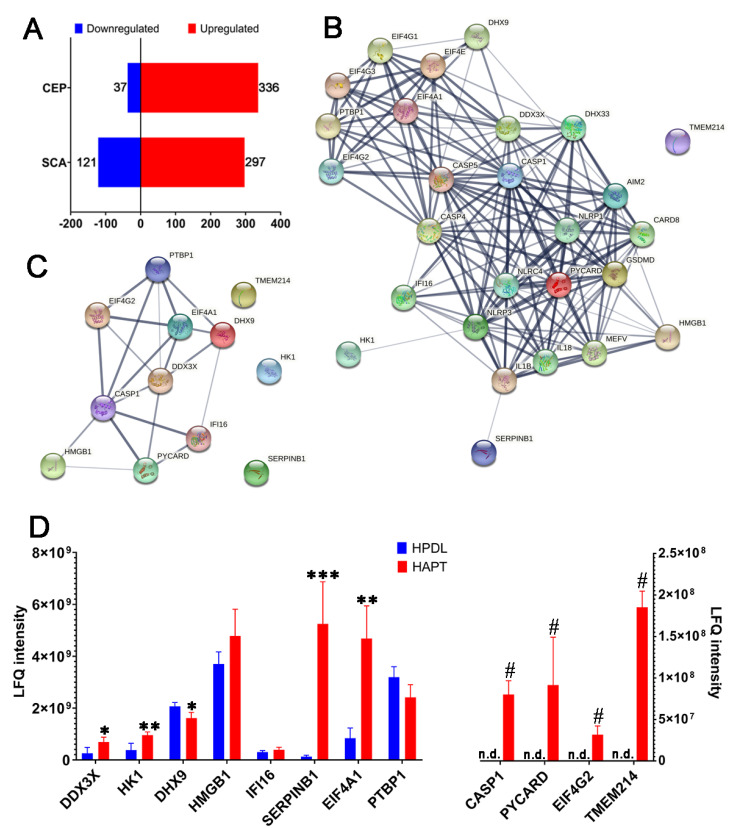
Caspase-1 and pyroptosis-related proteins in human apical periodontitis tissue (HAPT). (**A**) The number of SCA and CEP in HAPT and periodontal ligament tissue (HPDL). (**B**) The PPI network including all proteins possibly related to pyroptosis according to STRING. (**C**) The PPI network including proteins related to pyroptosis in our study. Caspase-1 is related to most other proteins. (**D**) The LFQ intensity of 12 proteins related to pyroptosis in our study was detected by label-free proteomics (*, *p* < 0.05; **, *p* < 0.01; ***, *p* < 0.001; #, proteins only detected in HAPT; n.d., not detected).

**Figure 2 biomolecules-12-01660-f002:**
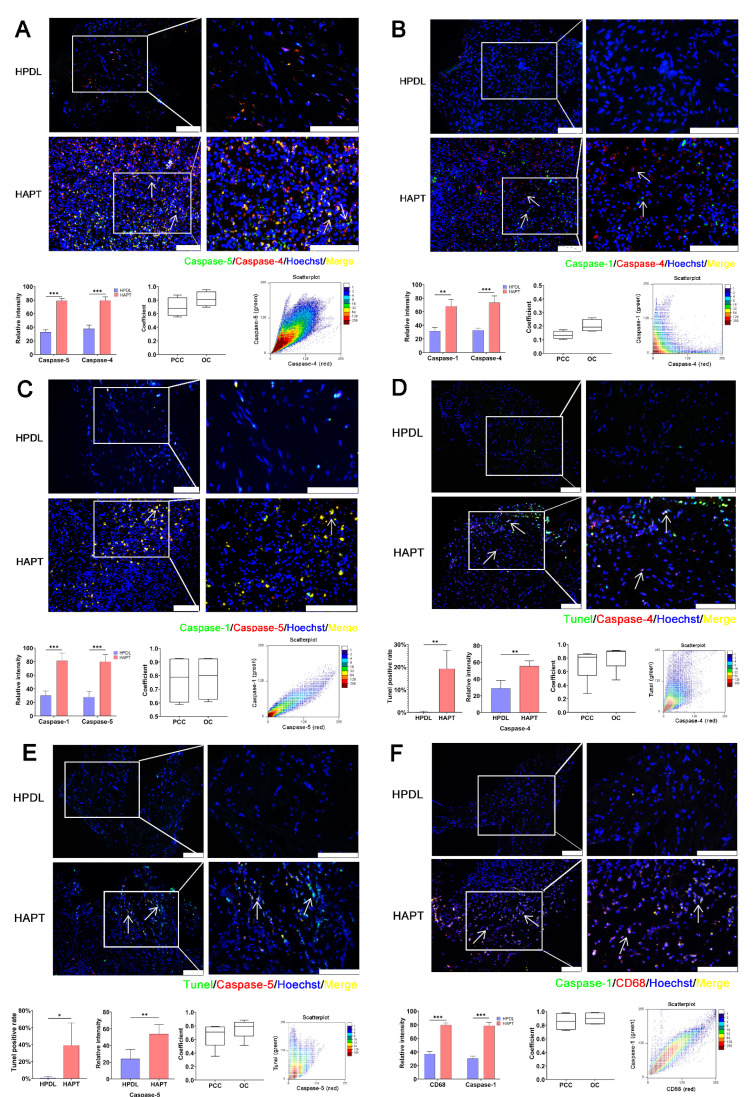
Caspase-1/-4/-5 mediate different levels of pyroptosis in human AP. (**A**) Triple labelling of caspase-5 (green), caspase-4 (red) and Hoechst (blue) was performed by using immunofluorescence. Caspase-4 and caspase-5 partially overlapped. (**B**,**C**) Triple labelling of caspase-1 (green), caspase-4/-5 (red) and Hoechst (blue) was performed by using immunofluorescence. Caspase-1 and caspase-4 did not overlap, but caspase-1 and caspase-5 overlapped. (**D**,**E**) TUNEL (green), caspase-4/-5 (red) and Hoechst (blue) staining were performed using TUNEL staining and immunofluorescence. Most of the caspase-4/-5^+^ cells underwent pyroptotic cell death (Bar, 100 μm). (**F**) Triple labelling of caspase-1 (green), CD68 (red) and Hoechst (blue) was performed by immunofluorescence. The results showed that most CD68^+^ macrophages expressed caspase-1 (Bar, 100 μm). (**G,H**) Triple labelling of CD68 (green), caspase-4/-5 (red) and Hoechst (blue) was performed by immunofluorescence. The results indicated that caspase-4 was not activated but caspase-5 was activated in CD68^+^ macrophages (Bar, 100 μm). *, *p* < 0.05; **, *p* < 0.01; ***, *p* < 0.001.

**Figure 3 biomolecules-12-01660-f003:**
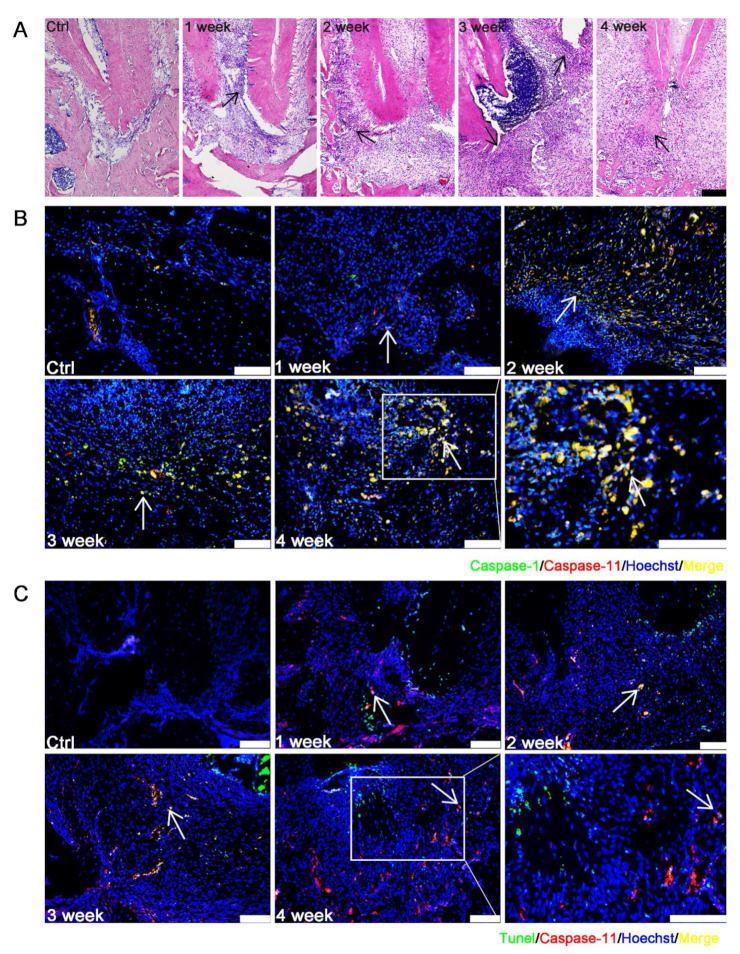
Pyroptosis was detected in EAP. (**A**) The HE staining of EAP. The apical lesion developed gradually (Bar, 100 μm). (**B**) Triple labelling of caspase-1 (green), caspase-11 (red) and Hoechst (blue) was performed by using immunofluorescence. The results showed that caspase-1 and caspase-11 overlapped (Bar, 100 μm). (**C**) TUNEL (green), caspase-11 (red) and Hoechst (blue) staining were performed using TUNEL staining and immunofluorescence. Dual labelling of TUNEL and caspase-11 showed that some caspase-11^+^ cells underwent pyroptosis (Bar, 100 μm). (**D**,**E**) Triple labelling of caspase-1/-11 (green), CD68 (red) and Hoechst (blue) was performed using immunofluorescence. The results showed that caspase-1 and caspase-11 were expressed in AP, especially in macrophages (Bar, 100 μm).

**Figure 4 biomolecules-12-01660-f004:**
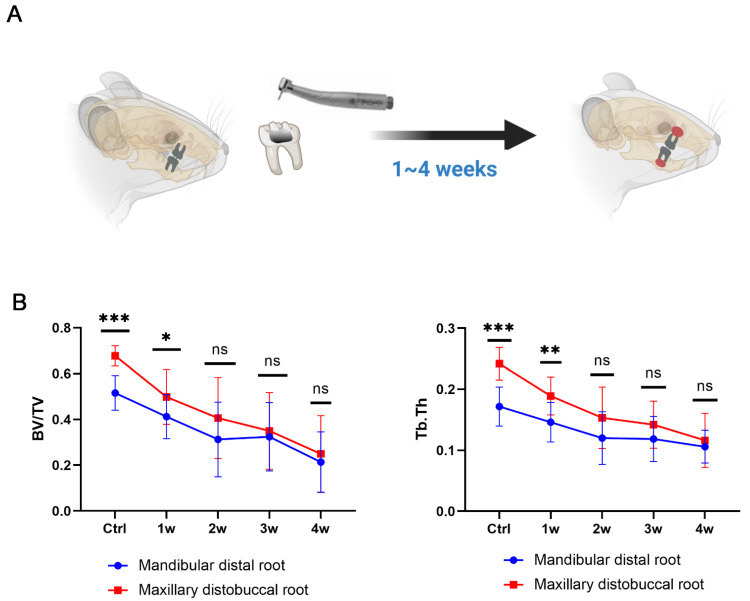
A dual EAP model was established with smaller mandibular lesions and larger maxillary lesions. (**A**) Establishment of the dual EAP model. (**B**) Changes in bone density in the maxillary distobuccal root and mandibular distal root (*n* = 14~16, *, *p* < 0.05; **, *p* < 0.01; ***, *p* < 0.001; ns, no significance).

**Figure 5 biomolecules-12-01660-f005:**
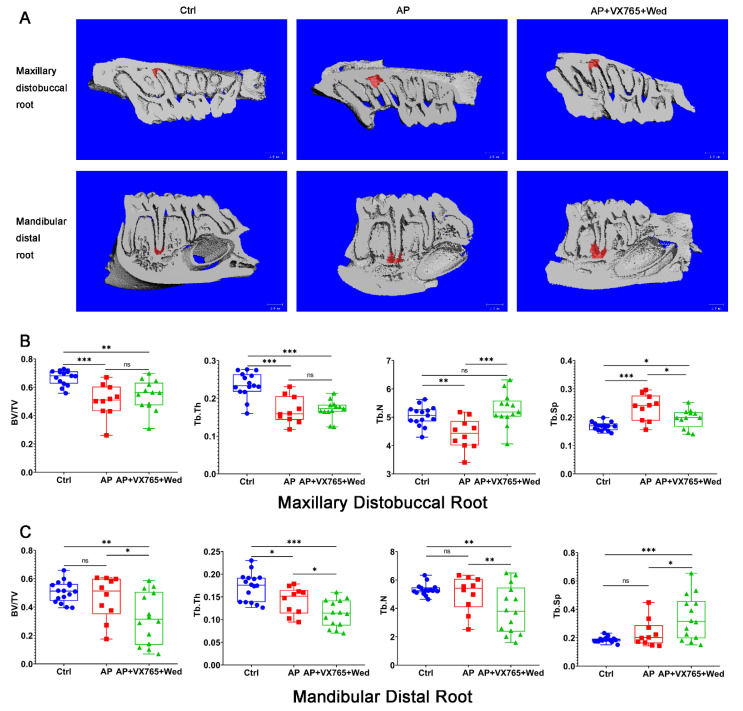
Inhibition of caspase-1 and -11 decreased bone loss when AP was larger (maxilla). In contrast, bone loss was increased when AP was smaller (mandible). (**A**) Three-dimensional reconstruction of EAP. (**B**,**C**) Bone loss of the maxillary distobuccal root and mandibular distal root (one-way ANOVA, *n* = 10~16; *, *p* < 0.05; **, *p* < 0.01; ***, *p* < 0.001; ns, no significance).

**Figure 6 biomolecules-12-01660-f006:**
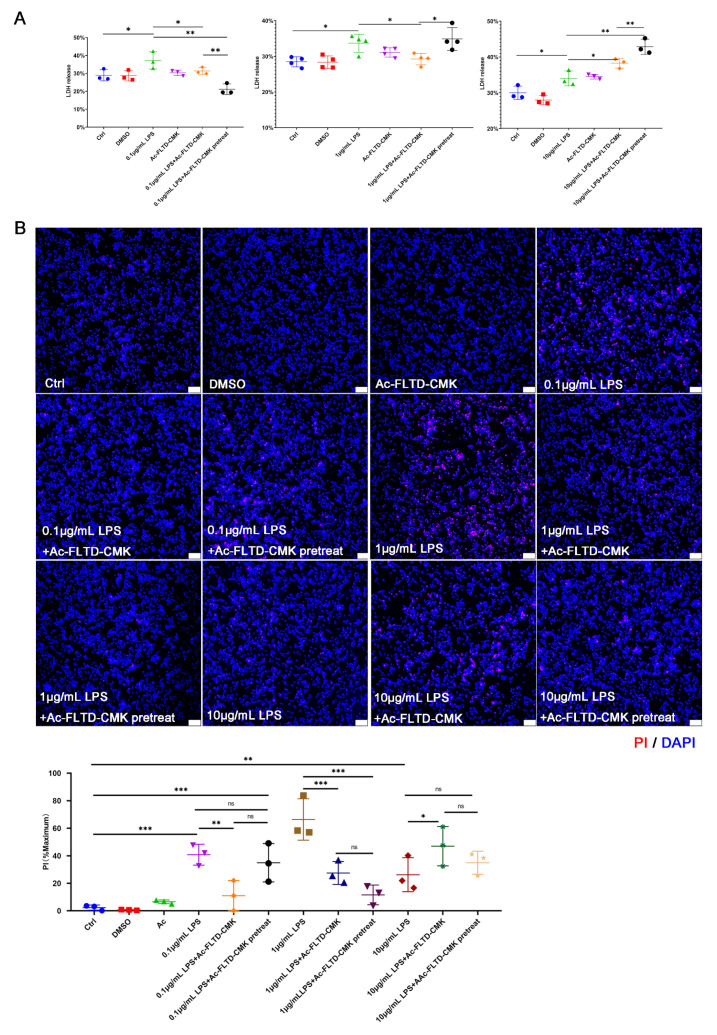
LDH release and PI staining in macrophages stimulated with different levels of LPS and different inhibition durations of Ac-FLTD-CMK. THP-derived macrophages were serum-starved and then stimulated with 0.1 μg/mL~10 μg/mL LPS for 6 h, with no pretreatment or pretreatment with Ac-FLTD-CMK. (**A**) LDH release from THP cells was detected. (**B**) PI staining was quantified by ImageJ software. (one-way ANOVA, *n* = 3~4; *, *p* < 0.05, **, *p* < 0.01; ***, *p* < 0.001; ns, no significance).

**Figure 7 biomolecules-12-01660-f007:**
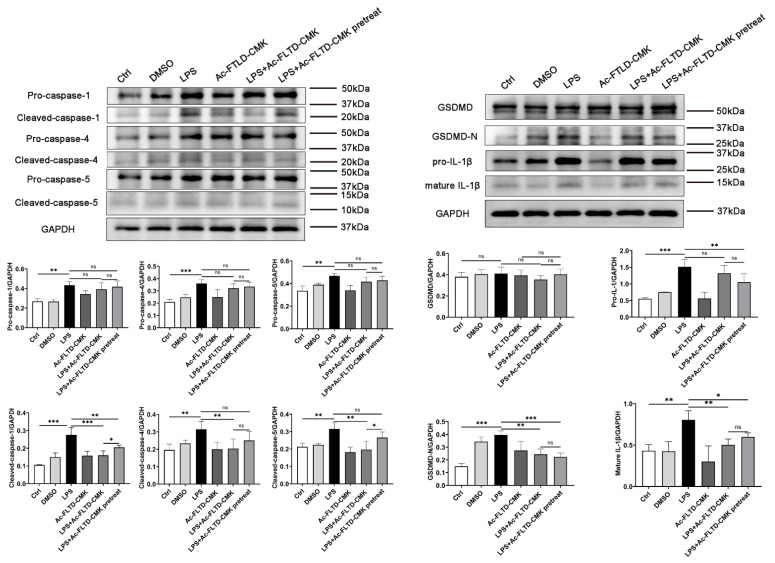
Caspase-1/-4/-5 were activated after LPS stimulation but inhibited by Ac-FLTD-CMK. THP-1-derived macrophages were stimulated with 1 μg/mL LPS for 6 h with or without Ac-FLTD-CMK. Pro-caspase-1, cleaved-caspase-1, pro-caspase-4, cleaved-caspase-4, pro-caspase-5, cleaved-caspase-5, GSDMD, GSDMD-N, pro-IL-1β and mature IL-1β were examined by Western blot. Data are represented as the mean ± S.D. (one-way ANOVA, *n* = 3; *, *p* < 0.05; **, *p* < 0.01; ***, *p* < 0.001; ns, no significance).

**Figure 8 biomolecules-12-01660-f008:**
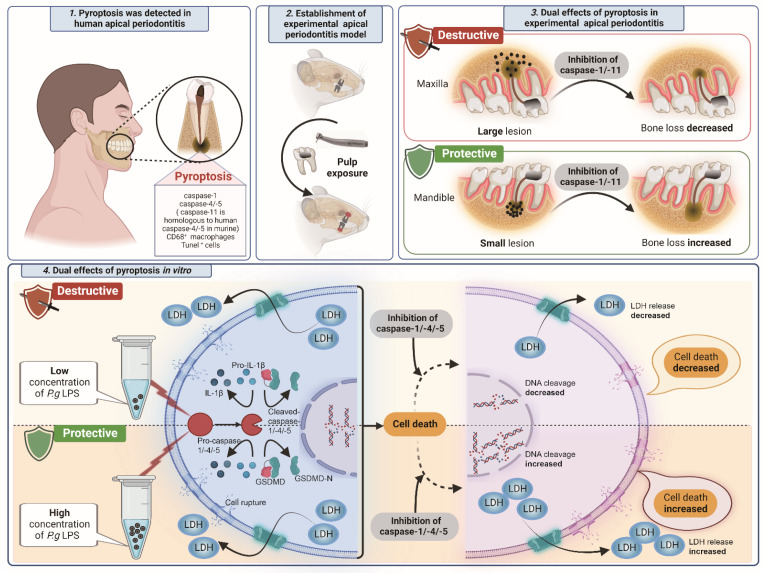
The schematic diagram of dual roles of pyroptosis in apical periodontitis.

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
