# Peer review of "Double-Edged Sword Effect of Pyroptosis: The Role of Caspase-1/-4/-5/-11 in Different Levels of Apical Periodontitis"

_biomolecules, 2022, doi:10.3390/biom12111660_

Round 1
Reviewer 1 Report
In this manuscript, Wu et al tried to clarify the effect of canonical and noncanonical pyroptosis in apical periodontitis, and found that double-edged sword effect of pyroptosis in apical periodontitis. The study contains potentially interesting findings, but data quality, especially image quality of immunofluorescence in most of Figures is too poor to be published. Several improvements are necessary to be accepted for publication in Biomolecules.
1. Abstract is confusing. It is not clear which part of analyses using clinical samples or mouse model.
2. In Figure 1A, it is better to show the list of identified proteins, at least in supplemental table.
3. In Figure 1D, it is indicated “HAP” but not in text part. It mostly indicated “HAPT” in main text. Whichever OK but should be unified through the manuscript.
4. In Figure 1D, although it is indicated “0 (zero?)”, it is not suitable there. It should be “not detected (n.d.)” or something like.
5. In Figure 2, 3, and 6, image quality of immunofluorescence is poor. Please replace high quality images, and better to add image of higher magnification.
6. In Figure 2A-C, results of scatterplot seemed not to be logical. In A, caspase 5 and 4 seemed to be correlated, caspase 1 and 4 seemed to be correlated in C. But, caspase 1 and 4 looked like correlating inversely in B. How do you explain this?
7. In Figure 3A, histological features should be indicated using arrow or somehow.
8. In Figure 5A, what does it mean “large lesion” and “small lesion”. Does it mean just size of lesions? Are they real images? Then, “large lesions” do not look like large, “small lesions” not small. Please explain details.
9. In Figure 5C, horizontal indication is hiding.
10. In Figure 6, they are interesting results, but it thus should be carefully examined. Sample size is too small. Evaluation of effects between 0.1 and 1μg/mL of LPS is recommended.
11. In Figure 7, indications around graphs are too small to see.
Author Response
Thank you very much for an excellent review and the helpful comments. The responses to your comments could be found in attachment.

Reviewer 2 Report
This manuscript describes that caspase-1/-4/-5/-11 has double-edged sword effects of pyroptosis in apical periodontitis by both in vitro and in vivo experiments. Overall, this manuscript is very interesting and relatively-good article. However, this manuscript needs to be addressed to some points as described below.
General comments:
Please describe the reason why the bone loss of rat was not determined after 5 weeks.
Materials and Methods in Appendix 1 represents that the rats were randomly divided into 5 groups. However, Figure 5 shows the results from only 3 groups. Please show all results from 5 groups.
Figure 5B and 5C does not show that the difference between “control” group and “AP+VX765+Wed”. Please present the results from statistical analysis between these groups.
In Figure 7, the level of GSDMD-N from DMSO group seems to be increased than that from control group. Please describe this reason.
All western blotting images shown in Appendix has no marker lane showing reference molecular weight. Please show the clear images with marker lane showing reference molecular weight.
All western blotting images shown in Appendix has nonspecific multiple bands. Therefore, the indicated bands in western blotting images seem to not be specific.
The band intensities between Figure 7 and images in Appendix were different.
The image of “Mature IL-1b” shown in Appendix is not full image. Please show this whole image.
Specific comments:
Page 1, Lines 3 and 4 from the last: Please change “gasermin” to “gasdermin”.
Page 2, Lines 8-9: Please change “interleukin (IL)-1b” to “IL-1b”.
Figure 8: Please change “1. Pyroptosis weas detected in” to “1. Pyroptosis was detected in”.
Author Response

(The authors gave the same response as above.)

Reviewer 3 Report
Dear Authors,
I have read and analyzed the manuscript ”Double-edged sword effect of pyroptosis: the role of Caspase- 1/-4/-5/-11 in different levels of apical periodontitis” and I want to congratulate you for your hard work and for your results!
The Introduction is well written, the Methods are adequately described and the Results are clear (maybe, Figures 6 and 7 could be a bit enlarged). Also, the Discussions and Conclusions are pertinent and in accordance to your results.
Overall, I consider the manuscript a valuable contribution to the field and suitable for publication.
Sincerely yours,
Author Response
Thank you very much for an excellent review and the helpful comments.
Round 2
Reviewer 2 Report
This manuscript has been improved for some suggested points.
However, several suggested points have not yet been modified.
Please present each molecular weight size (kDa) with arrow on reference marker line.
The image of “Mature IL-1b” shown in Appendix is not still full image. Please show this whole (full) image with reference marker line.
Author Response
Thank you very much for the outstanding review. Responses to your comments could be found in attachment.
